# A Salutogenic Approach to Disaster Recovery: The Case of the Lac-Mégantic Rail Disaster

**DOI:** 10.3390/ijerph17051463

**Published:** 2020-02-25

**Authors:** Mélissa Généreux, Mathieu Roy, Tracey O’Sullivan, Danielle Maltais

**Affiliations:** 1Faculty of Medicine and Health Sciences, Université de Sherbrooke, Sherbrooke, QC J1H5N4, Canada; 2Eastern Townships Public Health Department, Sherbrooke, QC J1H1R3, Canada; 3Health Technology and Social Services Assessment Unit, Eastern Townships Integrated University Health and Social Services Centre, Sherbrooke, QC J1H 4C4, Canada; Mathieu.roy7@usherbrooke.ca; 4Interdisciplinary School of Health Sciences, Faculty of Health Sciences, University of Ottawa, Ottawa, ON K1N 6N5, Canada; tosulliv@uottawa.ca; 5Department of Human and Social Sciences, Université du Québec à Chicoutimi, Saguenay, QC G7H2B1, Canada; Danielle_Maltais@uqac.ca

**Keywords:** disaster, psychosocial impacts, community resilience

## Abstract

In July 2013, a train carrying crude oil derailed in Lac-Mégantic (Canada). This disaster provoked a major fire, 47 deaths, the destruction of 44 buildings, a massive evacuation, and an unparalleled oil spill. Since 2013, Public Health has undertaken several actions to address this challenging situation, using both quantitative and qualitative methods. Community-based surveys were conducted in Lac-Mégantic in 2014, 2015 and 2018. The first two surveys showed persistent and widespread health needs. Inspired by a salutogenic approach, Public Health has shifted its focus from health protection to health promotion. In 2016, a Day of Reflection was organized during which a map of community assets and an action plan for the community recovery were co-constructed with local stakeholders. The creation of an Outreach Team is an important outcome of this collective reflection. This team aims to enhance resilience and adaptive capacity. Several promising initiatives arose from the action plan—all of which greatly contributed to mobilize the community. Interestingly, the 2018 survey suggests that the situation is now evolving positively. This case study stresses the importance of recognizing community members as assets, rather than victims, and seeking a better balance between health protection and health promotion approaches.

## 1. Motivation

### 1.1. The Lac-Mégantic Train Derailment Disaster

On 6 July 2013, in the middle of the night, a train carrying crude oil derailed in the heart of Lac-Mégantic (Quebec, QC, Canada). This small town, situated on a lakeshore in the Estrie region of the province, has a population of 6000 residents. 

The train, with no engineer at the controls, spontaneously rolled downhill from its night stop location 11 kilometres away, toward the town of Lac-Mégantic. With a relatively constant downhill slope, the train’s descent accelerated to almost 100 kph by the time the locomotives encountered a sharp curve in downtown Lac-Mégantic and most of the trailing cars derailed. As they derailed, 63 tank cars ruptured and escaping crude oil ignited, leading to a succession of powerful explosions and a major conflagration. The fire spread rapidly to nearby structures, destroying 44 buildings. The derailment, the explosions and the subsequent fire resulted in 47 deaths and necessitated mass evacuation of 2000 persons, equivalent to one-third of the town’s population. 

With the coupling of human suffering and environmental degradation, the Lac-Mégantic derailment caused serious psychosocial and economic consequences, including the relocation of many families forced to leave their homes, loss of many jobs, and closure of many local businesses for weeks before relocating elsewhere in town [1]. Given the impact of this technological disaster, the involvement of public health personnel and resources was critical throughout the emergency response operations. The Public Health Department (PHD) for the Estrie region responded immediately to provide direct services needed to protect the citizens of Lac-Mégantic and on-site responders from several health hazards. The priority at that time was to assess, communicate, and manage immediate risks to public health associated with exposure to chemical, physical and biological agents [1]. 

In the face of disasters, it is important to recognize that the operational domain of public health in affected communities extends beyond health protection and disease prevention to include promotion of health and well-being. It is with this in mind that Estrie PHD, in collaboration with researchers in the field of psychosocial recovery, conducted a population health survey entitled “Enquête de santé populationnelle estrienne” (ESPE) in 2014. Unknowingly, this was the first of a long series of promising initiatives to mobilize the local community in this post-disaster landscape.

### 1.2. The Salutogenic Approach

The approach and orientation of the Theory of Salutogenesis are now well established in health sciences and used in various health promotion settings and contexts [2]. Unlike traditional preventive approaches aimed at identifying risk factors, limitations or diseases, asset-based approaches are used to identify factors fostering well-being, resources, or abilities [3]. According to the scientific literature, a greater stock of health assets empowers individuals and communities and helps to improve health and well-being. This is true both directly (i.e., health assets are associated with better health outcomes [4]) and indirectly (i.e., health assets moderate the relationships between a disadvantaged social position and negative health outcomes [5,6]). Examining the role of salutogenic factors in disaster contexts, however, needs further exploration as the field of health emergency and disaster risk management places much more focus on hazards, risks factors, vulnerability, and short- and long-term adverse health outcomes.

### 1.3. Objectives

Through the case of the Lac-Mégantic train derailment tragedy, we aim to discuss how salutogenesis can be used as a relevant and complementary framework in disaster settings, and how it can be incorporated into post-disaster recovery strategies to promote resilience. More specifically, the objectives of this case study are to: (1) describe the salutogenic approach applied to the Lac-Mégantic train derailment, (2) present the long-term psychosocial outcomes of this disaster, and (3) discuss some benefits observed from applying a salutogenic approach in a post-disaster landscape.

## 2. Approach

### 2.1. Quantitative Approach in Addressing a Challenging Environment

Any collective trauma, including technological or natural disasters, is likely to lead to adverse health impacts among survivors and the wider community. Due to the experience of extensive stress and loss, people exposed to large-scale disasters like the Lac-Mégantic train derailment are subject to long-term adverse outcomes [1]. There is now a solid evidence base for the substantial effects of such a tragic event on psychological health in directly affected communities, which may persist over time in the absence of adequate support. Interestingly, disasters may also result in positive psychological outcomes in some exposed persons [7,8,9,10,11,12]. 

Among actions that can be performed by public health agencies to bring support to local communities following disasters, monitoring long-term psychosocial outcomes (both positive and negative ones) is certainly relevant. Monitoring helps tailor interventions aimed at supporting affected individuals and communities, by promoting their resilience and recovery processes. The Estrie PHD, in close collaboration with the “Université du Québec à Chicoutimi” (UQAC) and the “University of Sherbrooke”, has therefore spent the first years following the event tracking the health needs and assets of those living in the Granit area using repeated cross-sectional community-based surveys.

In 2014, one year following the rail disaster in Lac-Mégantic, the PHD conducted a first health survey using a community-based random sample of 811 adults from the Granit area and additional 8000 adults residing elsewhere in the Estrie region. This representative sample responded to a telephone or web survey covering a variety of physical and psychological health outcomes. The second phase of the ESPE was carried out in the fall of 2015 and sought to better understand the local population’s health and well-being, along with its possible link to the July 2013 railway disaster. In total, 1600 adults were recruited randomly in 2015 to take part in this large-scale telephone survey. These included 800 from the Granit area, and 800 from elsewhere in the Estrie region. In the fall of 2018, a third, similar, survey was conducted and is referred to as phase 3. Each of these three studies is composed of a separate sample of adults residing in the Granit area or elsewhere in Estrie; the original sample of participants was not monitored across time. While an additional study was conducted in 2016 by UQAC, a different sampling strategy was used; therefore, it is not used for comparison with the other surveys [13,14].

The adults who agreed to participate in these studies were asked to answer an anonymous questionnaire, which took approximately 30 min to complete. A number of questions were identical across all three surveys, allowing for the comparability of results over time (years 1 to 5 following the tragedy). Various psychosocial outcomes were examined, including adverse effects of disasters (e.g., psychological distress, depressive episodes, signs of post-traumatic stress, diagnosed anxiety or mood disorders, social worker or psychologist consultation, anxiolytic drug use, alcohol abuse), but also positive ones (e.g., resilience, positive mental health, sense of coherence, sense of community belonging, social support). The following outcomes, all self-reported, were examined in at least one of the three cross-sectional surveys. These outcomes have been described more thoroughly elsewhere [8,15].

Deficit-based outcomes: Perception of fair or poor general health, excessive drinking episodes (at least once a week), finding most of the days stressful, psychological distress in the past month, based on the 6 item Kessler Scale (K6, ≥7; [16]), signs of post-traumatic stress in the last week (specific to the train derailment) based on the 15 item Impact of Event Scale (score ≥ 26 [17,18]), diagnosed anxiety disorders, diagnosed mood disorders, social worker or psychologist consultation in the past year, and perception of insecurity in the neighbourhood.

Asset-based outcomes: Resilience in the past month, based on the 10 item Connor–Davidson resilience scale (score ≥ 30; [19]), positive mental health, in the past month, based on the 14 item Mental Health Continuum-Short Form questionnaire [20,21], sense of community belonging, sense of coherence, based on the short version (3 items) of the sense of coherence (score ≥ 5; [22]), social support, based on the Multidimensional Scale of Perceived Support (score ≥ 69; [23]).

### 2.2. Qualitative Approach in Addressing a Challenging Environment

The release of the ESPE 2015 data (i.e., in February 2016) stimulated the emergence of health promotion and advocacy interventions for the local population in Lac-Mégantic. Given the magnitude of the tragedy, it was necessary to take a step back to understand the situation in relation to the normal process of community recovery. It was in this context that in March 2016, the Estrie PHD intensified its work with community partners, first by organizing a day of collective reflection. The purpose of this initiative was to work together to gain understanding of the situation and reverse the cycle. During this day, no fewer than 50 key actors (decision makers, stakeholders, citizens and experts) gathered. The reflection day was divided into two parts: (1) conference and workshops on resilience and lessons learned from the past and (2) conference and workshops on levers for long-term recovery and priorities for the future. 

A defining moment during the Day of Reflection occurred during an asset-mapping activity through which participants were invited to construct together an historical timeline that traces key milestones in recovery of their community and to recognize the progress made (Figure 1). More precisely, they were first divided in subgroups, where they had to highlight good moves, or successful interventions and initiatives implemented by local partners and citizens since the tragedy. Then, subgroups had to share their respective thoughts to the larger group in order to collectively construct the timeline. By doing so, the large group was able to identify a wide and diversified range of local assets, including physical, cultural, economic, social and spiritual ones, that all created positive effects on the community.

Throughout the Day of Reflection, a common vision of the desired future emerged and priorities for action and research were identified, leading to the co-construction of what would become the “Plan for the Recovery and Development of a Healthy Community in Lac-Mégantic and the Granit area”. This plan pursues the following objectives:Maintain and adapt psychosocial services to the needs of individuals and the community (outreach services),Stay connected with the community, andPromote community involvement.

In the weeks following the elaboration of the plan (i.e., April 2016), PHD advocated for additional funding to support its implementation. In June 2016, the “Ministère de la Santé et des Services sociaux” and the Canadian Red Cross announced substantial investments that would serve as financial levers to implement the adopted action plan. The ESPE data was an important contribution supporting an informed decision, based on understanding of the long-term psychosocial impacts of the tragedy.

In sum, holding such a Day of Reflection, which brought together key players from the community, contributed to the development of a common vision of solutions and the transmission of a clear, coherent and positive message to decision-makers and the community. 


*“Building a project together is really motivating. Especially since everyone feels involved: from citizens to elected officials. It was a very inspiring day!”*
—A participant of the collective reflection day.

This positive experience supports existing knowledge that beyond traditional surveys, qualitative methods are valuable for listening to, learning from, and engaging local partners and high-risk citizens. Through inclusive and empowering approaches, public health practitioners and researchers can better integrate members of the community as assets rather than victims and take into considerations their capacities in addition to their needs [24]. 

## 3. Results

### 3.1. Observations from the Community-Based Surveys (Quantitative Approach)

#### 3.1.1. The First Years Following the Disaster

In 2014, some differences were observed in the prevalence of deficit-based and asset-based psychosocial outcomes as a function of residential location (Lac-Mégantic, elsewhere in the Granit area, or elsewhere in the Estrie region) (Table 1). Many of these “psychosocial gradients” were stronger in 2015 [15]. Anxiety disorders, for instance, were twice as high in Lac-Mégantic residents as in other residents of the Estrie region in 2015 (14.1% vs. 7.2%, *p* = 0.003). In the same vein, adults in Lac-Mégantic, as opposed to those living elsewhere in the Estrie region, were less likely to report a high level of resilience in 2015 (47.8% vs. 63.3%, *p* < 0.0005), while this was not the case the year before. Similar observations were made for optimal mental health. 

Significant time trends from year one to year two post-disaster were also observed. While most psychosocial outcomes did not show any statistically significant improvement among adults, the use of psychosocial services decreased by half among adults residing in Lac-Mégantic between 2014 and 2015 [15]. 

Some deficit-based (e.g., post-traumatic stress) and asset-based outcomes (e.g., sense of coherence) were only examined as from 2015 (Table 2). Findings from the second wave revealed seven in ten adults in Lac-Mégantic showed moderate to severe signs of post-traumatic stress two years after the disaster. On another note, a strong sense of coherence was observed among 48.2% of adults residing in Lac-Mégantic, regardless of residential location, and this proportion was significantly lower than that observed elsewhere in the Estrie region (61.1%). These findings suggest that the stock of health assets can weaken with time among people directly impacted by a disaster, especially in the absence of adequate support and services [8].

#### 3.1.2. Long-Term Trends in Psychosocial Outcomes Following the Disaster

Given the increased efforts to support recovery in Lac-Mégantic in recent years, has there been any progress in terms of psychosocial outcomes? With regards to all the data collected from our three surveys, major findings emerge. First, the adverse psychosocial impacts observed in the years following the Lac-Mégantic rail tragedy in 2013 seem to be receding. For example, after reaching a peak in 2015, the proportions of adults reporting an anxiety disorder diagnosed by a doctor stabilized in 2018 in Lac-Mégantic. On the other hand, these proportions increased significantly elsewhere in Estrie from 2014 to 2018. In other words, the gap that had developed between Lac-Mégantic and the rest of Estrie in the first two years after the tragedy is no longer, in many respects [25].

Second, there was still a very high prevalence of signs of post-traumatic stress in 2018 (71.9%). Despite a gradual adaptation of citizens to the losses and stressors experienced during and after the 2013 tragedy, the local community has been deeply affected by the traumatic event and its aftermath. These markers could persist for many years, despite an outward appearance of adaptive functioning of individuals and their community. Finally, protective factors are increasingly observed in Lac-Mégantic, particularly social support and sense of belonging to the community that were especially strong in 2018 [25]. These factors may act as powerful moderators of the adverse effects of primary and secondary stressors typically arising from large-scale disasters.

### 3.2. Observations from the Field (Qualitative Approach)

#### 3.2.1. The Outreach Team

Following the day of reflection, in 2016, Estrie PHD created a permanent community Outreach Team in Lac-Mégantic. Located outside formal clinical settings (i.e., in the downtown area), this multidisciplinary team has focused on bringing psychosocial services closer to the population. Four full-time professionals (two social workers, one outreach worker and two community organizers), and two part-time professionals (a kinesiologist and a nutritionist) comprise the team. 

The following principles guided the entire Lac-Mégantic outreach initiative: global health, prevention, scientific rigour, a strengths-based approach, empowerment, inter-organizational and intersectoral collaboration, and inclusion. Citizen participation and community development were at the heart of this approach. A wide range of services are offered, ranging from daily interactions with citizens and local organizations (in the form of psychosocial support, response to service requests, rapid detection and response to emerging needs, collaboration with the organization of activities, etc.) to involvement in various projects emerging from the action plan [25].

#### 3.2.2. Promising Initiatives to Mobilize the Local Community

The EnRiCH (Enhancing Resilience and Capacity for Health) Community Resilience Framework for High-Risk Populations [24] inspired the strategies developed within this community to promote community resilience, health and well-being [26,27]. Based on qualitative research conducted in five Canadian communities and a review of scientific literature, this framework provides an asset-based integrated upstream and downstream approach to disaster risk. With the development and use of adaptive capacities as a central element, it advocates three pillars and four areas of intervention, as described in Table 3, all in a cultural context and working with the complexity specific to disasters.

In line with this reference framework, several promising initiatives have been implemented in recent years within the Lac-Mégantic community to activate community resilience, social cohesion and citizen participation in a post-disaster setting. Committed to keeping track of local innovations and sharing them in formats that are suitable for both experts and practitioners, a synthesis of some of these promising initiatives has been produced and updated on an annual basis by the Outreach Team since 2017 [28]. These initiatives (e.g., social animation, Photovoice, Greeters, walking club) all contributed significantly to empowering citizens and mobilizing the community of Lac-Mégantic and surrounding areas. These initiatives also appear to have had a positive impact on the mental health and well-being of the citizens of this community.

As is generally known, organizing community projects or collective events, increasing opportunities to become involved as citizens, as well as other elements that strengthen social capital, contribute to building resilience in a post-disaster context. The data collected in this regard from 800 adults in MRC du Granit in the framework of ESPE 2018 provides additional support for this knowledge (Figure 2 [25]).

##### Photovoice

In 2017, in collaboration with the University of Ottawa EnRiCH research team and PHD of Estrie, the citizens of Lac-Mégantic took part in a Photovoice Initiative to map the assets of their community and develop a positive campaign and vision for the community looking forward to 2025. Over a 6 month, period the Lac-Megantic Photovoice Group met monthly to take photos of community assets and ideas to support their vision for the community. They met to discuss their photos with the group and share their ideas around issues that matter to them. The Lac-Mégantic Photovoice Group planned and hosted two exhibitions to facilitate knowledge mobilization and foster dialogue with decision-makers in Lac-Mégantic and Ottawa, including local and federal politicians. The Photovoice Initiative was highlighted as an inspirational example of community engagement in resilience initiatives in a report by the World Health Organization [29].


*“We could express our sadness, our emotions openly because we were welcomed, without criticism. At first it was quite emotional, but over the meetings, this overflow was transformed into something lighter. It did me good. It made a big difference.”*
—A participant of the Photovoice Initiative.

##### Ephemeral Place

The population is struggling to reclaim the downtown area of Lac-Mégantic, which was largely destroyed during the railway tragedy of 2013. Being under reconstruction, this new place, full of meaning and memories, is a constant reminder of loss. At the same time, there is a desire among citizens to get involved and to revitalize their living environment. In 2018, Ephemeral Place in the heart of the city was created in response to this desire; it is a space to promote social activities, networking and gatherings. This outdoor venue, under the responsibility of the Outreach Team, encourages the involvement of citizens of all ages and all horizons to foster social participation. Since these are temporary installations, it is an opportunity to experiment with concepts or ideas, while creating positive experiences. Through its free and varied leisure activities offered to citizens (5 to 7 with musicians, barbecues, outdoor film screening with popcorn, laughter yoga, intergenerational karaoke, etc.) and its unique approach, Ephemeral Place undeniably supports the long-term recovery of the Lac-Mégantic community [30].

##### Lessons Learned from a Citizen’s Perspective

Inspired from a similar initiative following the bushfires which swept across Victoria, Australia, in 2009 [31], the idea behind this project was to collect statements from people who experienced the tragedy through one-on-one and/or group interviews and to identify overriding themes. Driven by the Outreach Team, this initiative provided a voice and brought together people who wished to contribute in this way, naming what could be changed or improved as a way of managing future disasters. Through their experience, citizens could make recommendations to the different bodies with which they interacted during the rail tragedy of July 2013 but also during the months and years that followed it. A semi-structured interview guide was developed, based on the CHAMPSS Functional Capabilities Framework [32]. The acronym stands for the following categories of functional capabilities: Communication, Awareness, Mobility/Transportation, Psychosocial, Self-Care and Daily Tasks, and Safety and Security. Approximately 12 interviews were conducted with citizens that would not have been reached otherwise, in order to make their voice heard. Data collected through these interviews was then pooled and analyzed to draw emerging themes that were sent to participants for a further validation. By being inclusive and recognizing the various experience lived, this project gave citizens a different opportunity to contribute following the tragedy. All this led to the writing of a document sharing post-disaster good practices, according to the perspectives of citizens with a unique field expertise in the matter. This document could then be submitted to the bodies concerned, upon approval from the group.

## 4. Recommendations

To our knowledge, this study is among the first to report how a salutogenic approach can contribute to improve health and well-being in the aftermath of a disaster. Only few studies used positive approaches in such type of setting. Leadbeater and colleagues [33] described how community leadership facilitated the social recovery process in the community of Strathewen (Australia), following the 2009 Victorian bushfires. Van Kessel [34], for its part, explained how promoting resilience mitigated impact on mental health after the 2010/11 Victorian floods and the 2009 Victorian bushfires. It is noteworthy that a recent systematic review highlighted a gap in the evidence relating to specific interventions targeting the resilience of adults who have experienced a disaster. Authors from this review call for more studies exploring the ability of interventions to build the intrinsic capacity of a community to adapt to disasters [35]. Despite the paucity of “real-world research” and knowledge on effective asset-based approaches in a post-disaster landscape, many theoretical or conceptual papers support the key role of community resilience in promoting population health in such settings [36,37,38,39,40,41,42]. 

In our case study, the objective was to discuss how a salutogenic approach was used to help the community of Lac-Mégantic in its recovery from a profound tragedy. Our various community-based surveys, combined with continuous on-the-ground presence of the PHD, have provided situational awareness about how the psychosocial impacts resulting from the 2013 rail tragedy decreased over time. Although this tragedy has left its mark, the local community is gradually adapting to its new reality. The asset-based approach used in recovery seems to have contributed to this “new reality” and emphasizes the importance of social capital to activate individual and community resilience in post-disaster contexts.

Many lessons have been identified from this unique and informative experience. First, long-term monitoring of psychosocial impacts through repeated community-based surveys is relevant, if not essential. Such surveys serve as powerful tools for health promotion initiatives and advocacy on behalf of the local population. Such survey support priority setting (e.g., targeting most at-risk populations) and promote risk-informed decision making. 

Second, the voices of various groups, including those at heightened-risk, should be heard to take account of their specific needs and capacities. It is important to take time to listen and learn from citizens and consider all members of the community as assets rather than victims. This is critical to promote concrete social measures and psychosocial support tailored to their needs.

Third, regardless of the extent of the problems observed in the field, public health must seek a balance between a health protection (focused on hazards and risk factors) and health promotion (focused on protective factors, local strengths and resources). 

Fourth, public health practitioners, academics and leaders must collaborate closely with local organizations and citizen groups. This is fundamental for a successful recovery. Putting citizens at the heart of all considerations helps to make sense out of a chaotic situation and contribute to the recovery of the community. 

Fifth, public health organizations should capitalize on existing knowledge to develop and apply strategies and interventions in a post-disaster context. As part of their recovery operations, they should also share their own knowledge and experiences (e.g., lessons learned, tools and resources).

Finally, this rich experience in Granit over the last six years enabled us to identify three key success factors in supporting the psychosocial recovery and social reconstruction following a disaster: Acknowledge the strengths of the community and promote citizen participation;A strong political commitment is essential to support the community through preventive actions, upstream of problems;A public health team must have the resources to be able to support the development and implementation of these actions.

## 5. Conclusions

This case study gives a concrete example of how asset-based approaches can be fruitful for enhancing community resilience and improving the health and well-being of a community in a post-disaster landscape. The positive evolution of the psychosocial situation in Lac-Mégantic, assessed both quantitatively and qualitatively, demonstrates the importance of developing a common understanding of risks and working together in finding solutions.

## 6. Future Work

Let us recall the importance of understanding, preventing and reducing psychosocial risks in the months and years following a disaster, whether natural, technological or intentional. In any case, concerted action to promote community resilience is required during, after, and ideally before the occurrence of such an event. As advocated by the United Nations, we must move from a disaster management logic to a risk management logic associated with these events, in partnership—rather than in silos—for the good of the community [43]. 

Disaster risk reduction, which is closely associated with climate change adaptation (due to the increasing number and intensity of recent disasters), is a pressing field of action for decision-makers, practitioners and researchers to promote health and well-being of communities and to increase their resilience for coping with multi-origin hazards.

While much is known about interventions targeting health needs following disasters (i.e., deficit-based models), less is known about what could foster resilience. Our case study was limited to a very simple design (e.g., post-disaster cross-sectional surveys and field observations). In order to generate stronger evidence-base intervention, future work in this research field should be based on high-quality studies (i.e., randomized or prospective cohort studies).

## Figures and Tables

**Figure 1 ijerph-17-01463-f001:**
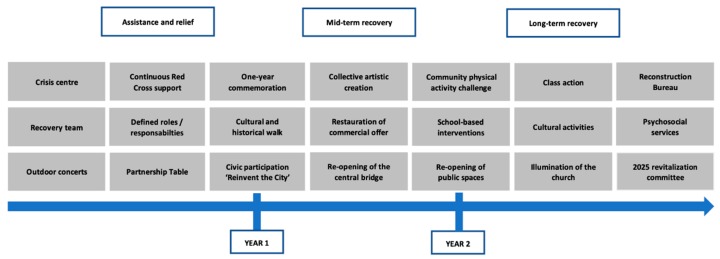
Historical timeline tracing key milestones in recovery of Lac-Mégantic community (March 2016).

**Figure 2 ijerph-17-01463-f002:**
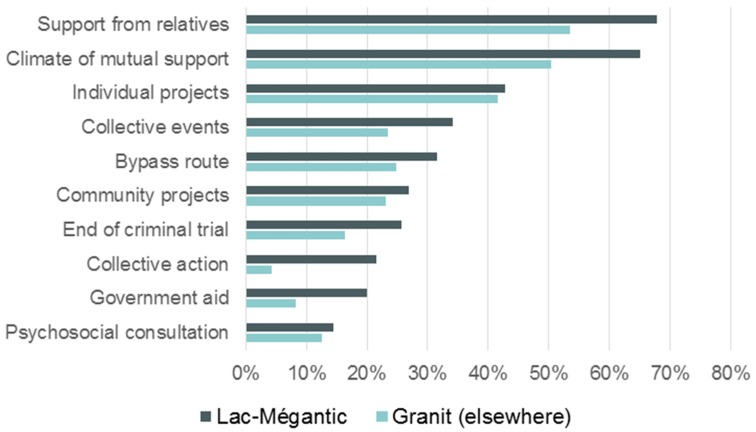
Elements that have significantly improved personal well-being over the past 12 months, “Enquête de santé populationnelle estrienne” (ESPE) 2018 (Granit area, 800 adults).

**Table 1 ijerph-17-01463-t001:** Deficit- and asset-based outcomes among a community-based sample of adults according to residential location, two years and five years post-disaster (e.g., Lac-Mégantic train derailment, 6 July 2013), Estrie region, 2014 and 2015.

	2014				2015			
	Lac-Mégantic	Granit (Elsewhere)	Estrie (Elsewhere)	*p* Value	Lac-Mégantic	Granit (Elsewhere)	Estrie (Elsewhere)	*p* Value
	*n* = 240	*n* = 571	*n* = 7926		*n* = 261	*n* = 539	*n* = 800	
**Deficit-based outcomes**								
Perception of fair/poor general health	13.0%	13.9%	12.9%	0.763	19.3%	9.7% (-)	9.6% (-)	<0.0005
Excessive drinking (≥1 episode/week)	10.5%	10.4%	10.2%	0.976	14.6%	13.3%	10.1%	0.087
Finding most of the days stressful	19.6%	21.2%	21.2%	0.862	24.9%	18.2%	22.2%	0.063
Psychological distress	28.9%	30.4%	23.8%	0.001	34.1%	23.2% (-)	22.1%	<0.0005
Anxiety disorder diagnosed by a physician	10.1%	7.0%	6.4%	0.080	14.1%	8.2%	7.2%	0.003
Mood disorder diagnosed by a physician	7.2%	5.5%	5.9%	0.621	9.4%	5.3%	6.6%	0.100
Social worker or psychologist consultation	26.9%	10.2%	10.3%	<0.0005	15.5% (-)	11.9%	11.7%	0.240
Perception of insecurity in the neighbourhood	8.2%	2.0%	2.5%	<0.0005	13.2%	2.8%	1.4%	<0.0005
**Asset-based outcomes**								
High level of resilience	55.9%	53.2%	56.0%	0.428	47.8%	55.8%	63.3% (+)	<0.0005
Optimal mental health	50.6%	47.0%	49.4%	0.504	44.5%	53.1% (+)	55.1% (+)	0.014
Strong sense of belonging to the community	80.5%	67.3%	55.9%	<0.0005	79.1%	78.2% (+)	67.0% (+)	<0.0005

**Table 2 ijerph-17-01463-t002:** Deficit- and asset-based outcomes among a community-based sample of adults according to residential location, two years and five years post-disaster (e.g., Lac-Mégantic train derailment, 6 July 2013), Estrie region, 2015 and 2018.

	2015				2018			
	Lac-Mégantic	Granit (Elsewhere)	Estrie (Elsewhere)	*p* Value	Lac-Mégantic	Granit (Elsewhere)	Estrie (Elsewhere)	*p* Value
	*n* = 261	*n* = 539	*n* = 800		*n* = 244	*n* = 564	*n* = 8022	
**Deficit-based outcomes**								
Perception of fair/poor general health	19.3%	9.7%	9.6%	<0.0005	11.3% (−)	12.3%	10.0%	0.196
Excessive drinking (≥1 episode/week)	14.6%	13.3%	10.1%	0.087	9.8%	9.3% (−)	7.6% (−)	0.177
Finding most of the days stressful	24.9%	18.2%	22.2%	0.063	23.8%	17.5%	20.3%	0.101
Psychological distress	34.1%	23.2%	22.1%	<0.0005	32.9%	26.5%	28.9% (+)	0.198
Moderate or severe post-traumatic stress	66.6%	35.1%	6.7%	<0.0005	71.9%	34.7%	N/A	<0.0005
Anxiety disorder diagnosed by a physician	14.1%	8.2%	7.2%	0.003	11.3%	8.4%	9.0%	0.351
Mood disorder diagnosed by a physician	9.4%	5.3%	6.6%	0.100	8.6%	5.5%	6.8%	0.252
Perception of insecurity in the neighbourhood	13.2%	2.8%	1.4%	<0.0005	3.2% (−)	1.5%	2.1%	0.238
**Asset-based outcomes**								
Strong sense of coherence	48.2%	47.9%	61.1%	<0.0005	46.6%	51.3%	54.8% (−)	0.039
Strong sense of belonging to the community	79.1%	78.2%	67.0%	<0.0005	81.8%	66.7% (−)	56.4% (−)	<0.0005
High level of social support	58.7%	67.9%	67.2%	0.033	67.6%	63.7%	67.3%	0.460

(+): Significant increase from 2015 to 2018. (−): Significant decrease from 2015 to 2018.

**Table 3 ijerph-17-01463-t003:** EnRiCH Framework components.

Components	Description
Adaptation capacity	Flexibility in changing environments
**Mainstay**	
Empowerment	Power to activate forces
Collaboration	Relationship with a common vision
Innovation	Emerging new practices
**Fields of intervention**	
Awareness and information	Collective sharing and learning
Strengths-based management	Mapping and linking forces
Upstream leadership	Proactive resource investing
Social connectivity	People and group networking
Complexity	Dynamic, non-linear context
Culture	Local community context

Source [24].

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
