# Peer review of "A Salutogenic Approach to Disaster Recovery: The Case of the Lac-Mégantic Rail Disaster"

_ijerph, 2020, doi:10.3390/ijerph17051463_

Round 1

Reviewer 1 Report

The paper deals with a very interesting topic, focusing on a salutogenic approach applied in a Canadian community. The authors provide a deep analysis of the approach used to identify factors fostering well-being, resources and abilities.

Overall, the paper is well designed and developed. However, minor revisions are suggested and they refer mainly to the discussion section that should include references to other studies dealing with the same topic.

Results should be discussed in the light of other studies/research applying the same approach or similar, highlighting also the cost and benefits of this approach compared with others.

References list should be enriched accordingly

Author Response

Response to reviewer 1

The paper deals with a very interesting topic, focusing on a salutogenic approach applied in a Canadian community. The authors provide a deep analysis of the approach used to identify factors fostering well-being, resources and abilities. Overall, the paper is well designed and developed.

1. However, minor revisions are suggested and they refer mainly to the discussion section that should include references to other studies dealing with the same topic.

To our knowledge, our case study is among the first to report how a global salutogenic approach can contribute in improving health and well-being following a catastrophic event. This is now more clearly specified in the manuscript.

Two “similar” studies have now been cited. These studies describe a) how community leadership facilitated the social recovery process in the community of Strathewen (Australia), following the 2009 Victorian bushfires, and b) how promoting resilience mitigated impact on mental health after the 2010/11 Victorian floods and the 2009 Victorian bushfires.

Leadbeater (2013). Community leadership in disaster recovery: a case study.

Van Kessel et al. (2014). Strategies to enhance resilience post-natural disaster: a qualitative study of experiences with Australian floods and fires.

We have also cited a recent systematic review that highlighted a gap in the evidence relating to specific interventions targeting the resilience of adults who have experienced a disaster. The same review calls for more studies exploring the ability of interventions to build the intrinsic capacity of a community to adapt to disasters.

Van Kessel et al. (2014). Resilience–Rhetoric to Reality: A Systematic Review of Intervention Studies After Disasters.

Despite the paucity of knowledge on positive approaches in a post-disaster landscape, we have added some theoretical or conceptual papers on the concept of community resilience in such settings, including:

Abramson et al. (2015). The Resilience Activation Framework: a Conceptual Model of How Access to Social Resources Promotes Adaptation and Rapid Recovery in Post-disaster Settings.

Castelden et al. (2011). Resilience thinking in health protection.

Gibbs et al. (2015). Community wellbeing: applications for a disaster context.

Jackson et al. (2017). Evidence for the value of health promotion interventions in natural disaster management.

Moreover, we now refer to three studies that have observed positive effects of various health assets (e.g. resilience, meaning, social cohesion and social support) in different disaster contexts (Deepwater Horizon Oil Spill, UK floods, and Peruvian earthquakes):

Aiena et al. (2016). Meaning, Resilience, and Traumatic Stress After the Deepwater Horizon Oil Spill: A Study of Mississippi Coastal Residents Seeking Mental Health Services

Greene et al. (2015). Resilience and Vulnerability to the Psychological Harm From Flooding: The Role of Social Cohesion.

Karlin et al. (2012). Social Support, Mood, and Resiliency Following a Peruvian Natural Disaster.

2. Results should be discussed in the light of other studies/research applying the same approach or similar, highlighting also the cost and benefits of this approach compared with others.

Please see response to comment #1

3. References list should be enriched accordingly.

Please see response to comment #1

Reviewer 2 Report

It is an interesting read, though the academic wring is very poor.

The authors are encouraged to improve their academic writing.

As it stands at the moment, the manuscript reads like a press release. 

Not clear on the methodology implemented, please provide a step by step explanation of how the work has been done.

The whole manuscript should be oranised into:

1 Motivation

2 Objectives

3 Methodology

4 Implementation of methodology

5 Results

6 Conclusions

7 Future Work

Please revisit: In the fall of 111 2018, a third, similar, survey was conducted and is referred to as phase 3. Each of these three studies 112 is composed of a separate sample of adults residing in the Granit area or elsewhere in Estrie; the 113 original sample of participants was not monitored across time. While a fourth study was conducted 114 in 2016 by UQAC, a different sampling strategy was used; therefore it is not used for comparison 115 with the other surveys [13-14]. ...

The 3rd survey was organised after the 4th survey - how did it happen?. If you cannot explain then, delete the last sentence about the 4th survey.

your bullet points add no value to the manuscript at all. They stand disconnected at the moment. Please use text instead.

Your Figure 1 needs explanation in the text. 

you jumped way too quickly to Results. You need to tell more about how you have applied your methodology.

Please replace "Discussion" for "Recommendations" or similar. The text does not read like discussion at all. 

Your conclusion seem disconnected from the rest of the material. Talking about things are were not mentioned in the main text. that is not acceptable.

Section on Future Work is missing.

Author Response

Response to reviewer 2

  1. It is an interesting read, though the academic wring is very poor. The authors are encouraged to improve their academic writing. As it stands at the moment, the manuscript reads like a press release. 

We are sorry to read this comment and we are not sure whether we can fully address it. Our manuscript is intended at reporting the various innovative approaches that have been developed and implemented locally after the train derailment, and its positive impacts on the community psychosocial recovery, all through a case study. Commonly used in social sciences, a case study is a research design that allows studying complex phenomena within their contexts, trough descriptive and exploratory analysis of a person, group or event.

We have previously published similar case studies about the public health response following the Lac-Mégantic disaster, and these papers have generated a large interest from the scientific community worldwide.

  • Généreux, M.; Petit, G.; Maltais, D.; Roy, M.; Simard, R.; Boivin, S.; et al. The public health response during and after the Lac-Mégantic train derailment tragedy: A case study. Disaster Health 2015, 3-4, 1-8.
  • Généreux, M.; Petit, G.; Roy, M.; Maltais, D.; O’Sullivan, T. The “Lac-Mégantic tragedy” seen through the lens of The EnRiCH Community Resilience Framework for High-Risk Populations. Can J Public Health 2018, 109, 261-267.

If you believe that the style of our paper is not suitable for this scientific journal, we suggest to submit it elsewhere, as it would not be reasonable to re-write it in order improve the academic writing, as suggested by the reviewer.

  1. Not clear on the methodology implemented, please provide a step by step explanation of how the work has been done.

Typically, the methodology section of a case study should include information about design, site, participants, methods of collecting data and steps in analyzing the evidence, which we think was done in the submitted manuscript, although less in an academic style as mentioned by the reviewer.

We realize that the structure proposed for this paper may not be perfectly suitable for a case study, as such a structure refers to a classic epidemiological study. To avoid any confusion, we therefore suggest to replace the section entitled “Methodology” for a section entitled “Approach” as in fact, we are not describing a methodology in the classic academic sense. We rather describe the local setting and context followed by the quantitative and qualitative approaches developed in Lac-Mégantic to better capture the local needs and assets, with the aim of tailoring public health interventions and hence better supporting community recovery.

The community-based health survey methodology has been described more in details elsewhere and the purpose of this case study was not to put too much emphasis on it, but rather to explain how various quantitative and qualitative approaches can be combined to improve our understanding of a complex, challenging and ever-evolving situation.

  1. The whole manuscript should be organised into:

1 Motivation

2 Objectives

3 Methodology

4 Implementation of methodology

5 Results

6 Conclusions

7 Future Work

Thank you for this suggestion. If the editor accepts this structure, we are happy to modify it as suggested. We also suggest to replace “Methodology” by “Approach” as mentioned above.

  1. Please revisit: In the fall of 2018, a third, similar, survey was conducted and is referred to as phase 3. Each of these three studies is composed of a separate sample of adults residing in the Granit area or elsewhere in Estrie; the original sample of participants was not monitored across time. While a fourth study was conducted in 2016 by UQAC, a different sampling strategy was used; therefore it is not used for comparison with the other surveys [13-14]. ...

The 3rd survey was organized after the 4th survey - how did it happen?. If you cannot explain then, delete the last sentence about the 4th survey.

We suggest to replace “fourth” by “additional” to avoid any confusion, as indeed this study was conducted between 2015 and 2018 and was therefore not the fourth one.

  1. Your bullet points add no value to the manuscript at all. They stand disconnected at the moment. Please use text instead.

We have removed the bullet points and used text instead.

  1. Your Figure 1 needs explanation in the text. 

The reviewer may not have seen the explanations about Figure 1 that were already in the text: “A defining moment during the Day of Reflection occurred during an asset mapping activity through which participants were invited to construct together an historical timeline that traces key milestones in recovery of their community and to recognize the progress made (Figure 1). By highlighting a series of interventions and initiatives previously implemented by social workers and other partners, the group was able to identify a wide and diversified range of local assets, including physical, cultural, economic, social and spiritual ones, that created positive effects.

We have nonetheless added some details about Figure 1, as requested by the reviewer.

  1. You jumped way too quickly to Results. You need to tell more about how you have applied your methodology.

Please see our response to comment #2.

  1. Please replace "Discussion" for "Recommendations" or similar. The text does not read like discussion at all. 

We have replaced “Discussion” for “Recommendations and conclusions” as suggested.

  1. Your conclusion seem disconnected from the rest of the material. Talking about things are were not mentioned in the main text. that is not acceptable.

We have replaced “Conclusions” for “Future work” as suggested.

  1. Section on Future Work is missing.

If such a section is acceptable by the editor, we propose to move the sentences initially located in the “Conclusions” section into this new section, as it opens on wider concepts, such as climate change and disaster-risk reduction.

Round 2

Reviewer 2 Report

The authors have improved the manuscript significantly. It is clear now what was intended and how the work was organised. 

I do accept all the changes proposed, mainly replacing Methodology for Approach sounds much better, etc.

The paper should have Conclusions. Please do not merge recommendations and conclusions. The reader needs to know clearly what the conclusions of your research are. Please do not use references in your Conclusions. It is about what you have found, not others.

The section on Future Work should clearly show how the research is going to continue. There should be a short discussion on the weaknesses of your research followed by strategies to overcome these weaknesses. And this is what the reader is expecting to see in Future Work. Honestly I do not see how Climate Change is related to your train derailment. Totally disconnected in my view.

Author Response

Response to reviewer 2

1.The authors have improved the manuscript significantly. It is clear now
what was intended and how the work was organised.

Thank you.

  1. I do accept all the changes proposed, mainly replacing Methodology for
    Approach sounds much better, etc.

Thank you.

  1. The paper should have Conclusions. Please do not merge recommendations
    and conclusions. The reader needs to know clearly what the conclusions
    of your research are. Please do not use references in your Conclusions.
    It is about what you have found, not others.

A short conclusion section was added as requested (without any reference).

4. The section on Future Work should clearly show how the research is going
to continue. There should be a short discussion on the weaknesses of
your research followed by strategies to overcome these weaknesses. And
this is what the reader is expecting to see in Future Work.

We have added a short discussion on the weaknesses of our research followed by strategies to overcome these weaknesses.

  1. Honestly I do not see how Climate Change is related to your train derailment.
    Totally disconnected in my view.

The relationship between climate change and disaster risk reduction is better explained now.